# Astrocytes and Tinnitus

**DOI:** 10.3390/brainsci14121213

**Published:** 2024-11-29

**Authors:** Paola Perin, Roberto Pizzala

**Affiliations:** 1Department of Brain and Behavioral Sciences, University of Pavia, 27100 Pavia, Italy; 2Department of Molecular Medicine, University of Pavia, 27100 Pavia, Italy; roberto.pizzala@unipv.it; 3Fondazione IRCCS Policlinico San Matteo, 27100 Pavia, Italy

**Keywords:** tinnitus, astrocytes, synaptic plasticity, neuroinflammation

## Abstract

Tinnitus is correlated with anomalies of neural plasticity and has been found to be affected by inflammatory status. The current theories on tinnitus, although still somewhat incomplete, are based on maladaptive plasticity mechanisms. Astrocytes play a major role in both neural responses to inflammation and plasticity regulation; moreover, they have recently been discovered to encode “context” for neuronal circuits, which is similar to the “expectation” of Bayesian brain models. Therefore, this narrative review explores the possible and likely roles of astrocytes in the neural mechanisms leading to acute and chronic tinnitus.

## 1. Introduction

In spite of a large literature production, the neural bases of tinnitus are still incompletely understood [1]. Both animal and human studies have found anomalies in brain connectivity, structure, and activity in tinnitus subjects (see reviews from [2,3,4]), but results from different studies are often contradictory, and as yet there is no complete consensus on a human neural tinnitus “signature”. In animal models, several standard procedures are used to induce tinnitus, such as noise trauma and high-dose salicylate [5], but concurring hearing disturbances, such as hyperacusis and hearing loss, may act as confounding factors regarding what is being measured [5,6,7]; moreover, the measured functional brain changes correlated to tinnitus vary depending on the inducing procedure [8,9]. This suggests the presence of a major gap in our understanding of tinnitus.

## 2. Tinnitus Models

Currently, several theoretical models of tinnitus have been developed, which include homeostatic plasticity [10], central gain enhancement [11], thalamocortical dysrhythmia [12], frontostriatal gain disturbance [13], and sensory precision alteration [14]; for an overview of tinnitus models see also [15]. Although recent studies suggest that a simple central gain enhancement does not correlate with tinnitus [7,16], which could instead correlate with neural noise changes [7,15] no theory has so far been able to explain all tinnitus features (see discussions in [10,15]).

In particular, the hierarchical predictive models, which employ a theoretical framework used in other neuroscience fields, mostly developed for the cortex [17], can provide a mechanism explaining the presence of tinnitus without permanent hearing loss [14], but their usefulness is mostly limited by a lack of understanding of the biological correlates of theoretical processes (although some cellular candidates are emerging [18]).

In the current hierarchical predictive model, chronic tinnitus is induced by a transient decrease in sensory precision [14], meaning that neural discharge becomes less informative on the signal it carries because of a decrease in the sensory system transduction signal (e.g., TTS after noise trauma [19]) and/or an increase in spike discharge noise [15]. Decreased sensory precision allows aberrant top-down prior updating (although it is unclear why this does not always happen), which redefines default perception to the sound of tinnitus. The main limit of this theory is its abstract nature which points at perception without giving information regarding the correlation with neural activity. When measuring the latter, however, several problems emerge: after noise trauma, the tinnitus percept is observed before any changes in the spontaneous activity of the auditory system, which slowly emerge at a later time [10]. Moreover, tinnitus can be triggered by both cisplatin and carboplatin, but only the former induces firing rate increases in the inferior colliculus [10]. A change in astrocyte responses could explain the instant modification of neural responses without changes to the electrical discharge, since astrocytes are not electrically excitable [20] but have been found to modify neural responses with a subsecond timescale [21].

Tinnitus-correlated changes in the neural discharge intensity and synchronization are observed in several stations of the auditory CNS and in non-auditory structures; at the synaptic level, these changes are accompanied by a modification in both glutamatergic and GABA/glycinergic transmission [22,23]. Once again, astrocytes may be involved in these processes, since they are able to modulate transmission and plasticity at both glutamatergic and GABAergic synapses (reviewed in [24]; see also below).

## 3. Astrocytes

Although most studies on the neuroscience of tinnitus have focused on neuronal behaviour, it has been clear for several decades that glial cells, far from being an amorphous “glue” around neurons [25], play major roles in neural transmission as well [26]. In this perspective, we present evidence from auditory and other neuroscience fields that suggests that including astrocytes in the study of tinnitus mechanisms could help explain tinnitus mechanisms and features (Figure 1). This is based on three main points: (1) astrocytes are involved in neural plasticity and memory; (2) astrocytes are directly involved in inflammatory responses; and (3) astrocytes rapidly regulate local and global neuronal circuit responses, and their aberrant behaviour has been linked to several neural diseases. Since tinnitus is also associated with aberrant neural plasticity [27] and inflammation [28], exploring the possible roles of astrocytes in tinnitus could usher in novel therapeutical approaches for this condition. Other non-neuronal cell types, such as microglia [29,30] and oligodendrocytes [31], also play important roles in modulating neuronal response in the auditory system and tinnitus-related brain regions; however, they will not be the focus of this discussion. Similarly, the role of astrocytes in development will not be discussed here.

## 4. What Do Astrocytes Do?

Astrocytes are multitalented cells (Figure 2): in the adult brain, they play protective, metabolic, and computational roles [25,32]. The protection of the brain by astrocytes involves, among other things, organizing and maintaining the blood–brain barrier (reviewed in [33]), activating upon damage (in concert with microglia), with inflammation-related roles similar to those of innate immune cells (reviewed in [34]), and creating scars around lesion sites [35]. The metabolic roles of astrocytes include regulating local blood flow to match neuronal metabolic demand [36], transforming blood glucose in lactate for neuronal use [37,38], regulating thyroid hormones’ entry and actions within neural tissue [39], recycling neurotransmitters [24,40] and K+ ions [41], and gating the glymphatic clearance of the brain [42]. Finally, the computational roles of astrocytes include modulating synaptic transmission (the tripartite synapse model was the first glimpse of the role of astrocytes in neural computations [43]) and synaptic plasticity [44,45], sensing axonal conduction at Ranvier’s nodes [46], influencing the rhythmic activity of neuronal populations [47], and receiving and integrating salience signals into local neuronal circuit computations [48].

Although a single astrocyte is able to perform several of these roles, different specialized astrocyte subtypes have been known to exist for a long time (e.g., velate, fibrous, protoplasmic [50]), and more recent studies have classified astrocyte subtypes based on their shape, gene expression, and locations [45,51].

Within the auditory cortex [49], four astrocyte subtypes have been found, involved in axonal maintenance (AST1), synaptic transmission (AST2-3), and neurovascular coupling (AST4). Astrocytes are present throughout the auditory system, beginning with the central portion of the auditory nerve, up to the auditory cortex; they are, however, absent from the cochlea and spiral ganglion [52]. Interestingly, the cochlear nerve astrocytes extend further towards the periphery upon cochlear nerve compression [53] and proliferate in schwannoma [54] (which is associated with tinnitus [55]), and AxD mice, which have mutant GFAP in astrocytes, display more cochlear damage upon noise exposure [56]. In the cerebellum and hypothalamus, specialized astrocyte types named Bergmann glia [57] and tanycytes [58], respectively, are found in the layers contacting the brain surface. Interestingly, the dorsal cochlear nucleus (DCN) displays a neuronal paracerebellar organization [59], but lacks the synaptically-centered Bergmann glia, and instead displays tanycyte-like cells [60], which could receive chemical signals from the IV ventricle CSF or from the choroid plexus, which directly touches the DCN surface [61]. The DCN has been found in animal studies to be central for tinnitus onset [22]. In particular, DCN ablation abolishes tinnitus onset but not maintenance [62], and the balance between DCN and cerebellar circuits differs in animals developing and not developing tinnitus [63]. Although fusiform cell hyperactivity and/or hypersynchrony have been taken as the neural correlate of tinnitus in DCN [22,23], it is still unclear why this activity is only triggered in a fraction of noise-exposed animals. Since tanycytes can integrate multiple metabolic, neural, and endocrine factors, their activity in the DCN may be the key for understanding what triggers anomalous plasticity.

The study of astrocytes has lagged behind that of neurons because of their morphological complexity and dense interconnections and because their signals are difficult to measure (astrocytes lack electrical excitability [20] and instead generate intracellular Ca^2+^ signals [64]). Their shape is characterized by several branches emerging from the cell body (“astrocyte” means “star cell”), which terminate with several types of processes: AQP4+ endfeet contact blood vessels [65,66] and the brain surface [67], gap junction-containing terminals contact other astrocytes and different glial cells [68,69], thin terminal processes (PAPs, peripheral astrocytic processes) ensheathe neuronal synapses [70], and other branchlets contact neuronal axons at Ranvier’s nodes [71]. In addition, astrocytes can interact with surrounding cells through a diversity of chemical signals [72,73,74]. These connections form an astrocytic network which works in parallel to the neuronal one (Figure 3) [75,76].

Research on astrocytes has flourished recently owing to the availability of transgenic animal models in which their activation can be observed and modified in a controlled way [77]. In particular, mouse models based on optogenetic (reviewed in [77]) and DREADD (Designer Receptors Exclusively Activated by Designer Drugs, reviewed in [78]) allow for the selective modulation of astrocyte populations in vivo upon the delivery of light or drugs; moreover, fluorescent labeling allows for a quantification of astrocyte morphology and spatial patterning [79], Ca^2+^ signals [80], and of brain fluid pathways and barriers [81]. Finally, genes can only be selectively mutated in neurons or astrocytes (the gene Atm, involved in ataxia–telangiectasia, was found to affect cerebellar function only when mutated in astrocytes, but not in neurons [82]).

These and other animal models have greatly helped the understanding of communication between astrocytes and neurons; however, primate astrocytes display a larger size, complexity, and speed than those of rodents [83,84], and unique astrocyte types are present in the primate cerebral cortex [45]; therefore, the complexity of astrocyte functions in humans is likely higher than in rodents.

In clinical research, it also has to be kept in mind that different neuroimaging techniques monitor different cell populations. Whereas electrical measures of brain activity (such as EEG or MEG) directly measure neuronal responses, other techniques also reflect astrocyte properties [85]: fMRI BOLD signal, which reflects neurovascular coupling, may be affected by astrocyte activation even without changes in neuronal activity [86], and voxel-based morphometry, which quantifies differences in brain region volumes between pathological or physiological states [87], may variably reflect the volume changes of several cell types (including a prominent contribution by astrocytes) in humans [88] and animal models [89].

## 5. Astrocytes and Synaptic Transmission

Since the 1990s [43], it has been known that astrocytes are able to both sense and modulate synaptic transmission: they express receptors for most neurotransmitters and are in turn able to synthesize and release their own gliotransmitters, thus exerting direct feedback on neuronal activity [90]. While astrocytes are known to release several substances, the gliotransmitters most clearly identified are glutamate, d-serine, and ATP/adenosine, which modulate both excitatory and inhibitory synapses in many brain areas [91]. This modulation is very heterogeneous, because of several reasons. First, the same gliotransmitter can have opposite effects depending on the receptors expressed by the synapse: in the centromedial amygdala, astrocyte-derived ATP leads to the A1-mediated potentiation of inhibitory synapses and A2A-mediated depression of excitatory synapses [92]. Second, the effect of gliotransmission may combine with that of other transmitters and modulators: astrocyte-induced glutamate-mediated transient potentiation can become long-term potentiation when NO is released by the postsynaptic neuron [93]. Third, gliotransmission may affect adjacent synapses, as seen in endocannabinoid signaling [94]. Fourth, single astrocytes can release different gliotransmitters in response to different neuronal activity patterns, thus regulating transmission differently depending on the specific synaptic activation [95]. In addition, astrocytes surround and seal the synaptic cleft in a dynamic way, regulating how much transmitter is allowed to “spillover” to nearby synapses, in a way which is dependent on experience [96].

Although gliotransmission is now an established process, its complexity makes it difficult to extract astrocyte roles in neural computation. Depending on brain region [45], between 50% and 90% of synapses are contacted by astrocytes. A single astrocyte may contact up to 140,000 synapses in the rodent hippocampus [97], and up to 2 million in the human cortex [83], and at each synapse it may be stimulated by neurotransmitter release, thus locally increasing its cytoplasmic Ca^2+^ and/or modulating neurotransmission through gliotransmission [91]. Ca^2+^ signals in astrocytes are very complex, and still incompletely understood [64,80,98], but are known to include local signals triggered by several stimuli [99] and long-range signals that can rapidly propagate throughout several astrocytes through gap junctions [100,101]. Recent data show that when about ¼ of the astrocyte arborization has been activated, a Ca^2+^ surge occurs, consisting of an IP3R2-dependent Ca^2+^ wave spreading first to the astrocyte soma and then to the remaining arborization [102]. The meaning of these complex Ca^2+^ signals in neural computation is still unclear, but their features would allow astrocyte networks to perform complex computations [103] and modulate neuronal activity in a way that integrates local neural activity with immune [104], endocrine [105], and arousal [106] signals.

Classical neurophysiological experiments have described synaptic plasticity triggered by a co-activation of pre- and postsynaptic neuron activity [107]; however, plasticity may be modulated by a “reinforcement signal”, such as noradrenaline or dopamine [108]. The locus coeruleus (LC) is a noradrenergic nucleus involved in focusing processing on salient or goal-relevant information [108]. LC-derived noradrenaline is able to reorganize neural circuits by altering the strength of synaptic connections [109], and LC-derived noradrenergic effects target astrocytes [110]. Since tinnitus is correlated with changes in the salience circuits [3], this is one way astrocyte functions may directly affect tinnitus. In addition, since LC-derived noradrenaline also exerts an anti-inflammatory role [111], this circuit could provide the link between stress (which can damage LC neurons [112]), pathological attention (as in worry [113]), neuroinflammation, and tinnitus [28].

Since tinnitus has been associated with anomalies of inhibitory synaptic transmission in the auditory brainstem [23,114] and thalamus [115], it is pertinent to discuss the role of astrocytes in regulating these signals. Astrocytes are able to both release [91] and respond to [116] glycine, but their interaction with glycinergic synapses is not well characterized. As regards GABA, astrocytes are known to release it [117] and to sense its concentration through ionotropic and metabotropic GABA receptors and GAT1 and GAT3 transporters [118]. Unlike neurons, the activation of GABAA receptors is not inhibitory for astrocytes but may induce Ca^2+^ signals and/or gliotransmitter release [118]. This is due to the different regulation of Cl- gradient in astrocytes vs. neurons: adult neurons express the Cl transporter KCC2, which reduces intracellular Cl-, making Cl- current hyperpolarizing, whereas astrocytes express NKCC1, and therefore display higher intracellular Cl- and depolarizing Cl- currents [119]. In response to GABA, astrocytes may help neuronal inhibition by releasing Cl- in the extracellular space, which is then taken up by neurons. However, Cl- concentrations in astrocytes have been shown to be related to brain states (e.g., they are more stable during sleep and may vary within seconds after waking) [120]. Therefore, the modulation of GABAergic transmission by astrocytes may be strongly context dependent. Moreover, inflammatory signals such as bradykinin are able to directly affect astrocyte Cl- regulation by increasing VRAC currents [121]. VRAC Cl- channels are also involved in glutamate release from astrocytes under ischemic conditions and their activation may induce excitotoxicity [122].

After deafferentation [123] or hearing loss [124], the neuronal Cl- transporter KCC2 is downregulated in cochlear nuclei and inferior colliculus, reversing the inhibitory effect of GABA on neurons, similar to what is seen in the dorsal horn in neuropathic pain [125]. In these conditions, astrocyte modulation of extracellular Cl- (or, directly, of neuronal KCC2 expression [126]) could represent an important factor in the evolution of auditory brainstem responses, as seen in epilepsy [127]. Although deafferentation is clearly an inflammatory stimulus, which induces astrocyte and microglia proliferation [123], the complexity of synaptic interactions does not allow us to suggest cellular mechanisms linking deafferentation and tinnitus onset.

Long-term memory storage requires protein synthesis in neurons, and it is now possible to identify the “engram”, i.e., the neuron set involved in the storage of a given memory, by selectively labeling neurons that change their protein synthesis following a memory task [128]. In particular, hundreds of papers have shown changes in the transcription of IEGs (immediate–early genes) which are therefore commonly used as markers for this type of experiment [128]. Transgenic mouse lines are available where IEG expression activates the synthesis of a fluorescent protein [129], allowing direct observation of the engram with fluorescent imaging. A recent variant of this technique, called astrocyte-eGRASP [130], allowed the expression of fluorescent proteins not only in the engram neurons, but also in the astrocytes contacting them (Figure 3). With this technique, it has been observed that in the hippocampus, which is involved in episodic memory storage, astrocytes surrounding engram neurons display unique gene expression patterns, suggesting their involvement in engram maintenance [130].

The large number of connections of single astrocytes with synapses [83,97], together with the ability of astrocytes to communicate through gap junctions with each other and with other glial cells [68,72] and receive salience [110,131] and blood-derived signals [132] has suggested that these cells may represent the cellular correlate of context in neural circuits and give “contextual guidance” to neuronal networks [133]. In particular, cortical astrocytes may yield contextual modulation to the dendritic microcircuits which are thought to compute error [18].

Astrocytic Ca^2+^ waves can spread within processes and propagate to nearby astrocytes through gap junctions, producing a rhythmically pulsating network [134]. This mechanism is particularly important in the hippocampus, where it interacts with local electrical oscillations, modulating e.g., REM sleep [135] and in thalamocortical circuits, where it can modulate cortical gamma synchronization [136]. Thalamic astrocytes are also able to release GABA, thus modulating sensory acuity through increasing spike temporal precision in thalamocortical neurons [117]. The latter mechanism could yield a novel alternative basis for the noise increase postulated by the stochastic resonance model [15]: if thalamic astrocyte GABA release decreases, spike precision would also decrease, and noise would increase, without the need for a signal reduction in the periphery. GABA dysregulation in the auditory thalamus has actually been observed in tinnitus [115]. Moreover, thalamocortical dysrhythmia [12] has been theorized as a possible model of tinnitus onset. Further studies characterizing thalamic astrocyte function in tinnitus could shed more light on the issue.

## 6. Astrocytes and Neuroinflammation

The brain is referred to as an “immune privileged” organ, meaning that the presence of tissue grafts, pathogens, or cell death within the brain parenchyma does not induce a rapid infiltration of immune cells like that seen in other organs [137]. This is due to the presence of three barriers (the blood–brain barrier [33], the blood–CSF barrier [138], and the meninges with their immune populations [139]) and of a specialized lymphatic system, the glymphatic system, which starts in the brain parenchyma with astrocytes and connects to meningeal lymphatics [140]. The term “neuroinflammation” is used to describe several different processes, including the effects of systemic inflammation on the CNS, the inflammatory reaction of neural tissue, the alteration of glia in pathology, and the immune processes occurring in neural tissue [141]. Therefore, although astrocyte roles in neuroinflammation are undeniably central, the details of those roles need to be defined in a very clear way in order to be useful. In particular, for the auditory system, in different stations astrocyte roles are expected to differ, given their position. In particular, the DCN contains astrocytes exposed to the IV ventricle surface, whereas the VCN, inferior colliculus, MGN, and cortex contain glia limitans, and other brainstem nuclei have no access to the brainstem surface [142], therefore only containing BBB-related astrocytes. Specialized astrocytic populations have been observed in the auditory cortex [49] and DCN [60], and the former have been found to change upon salicylate-induced tinnitus [143]. In humans, tinnitus was correlated with a decrease in astrocytes in the inferior colliculus, together with a decrease in serotonergic and dopaminergic neurons from brainstem nuclei [144]. These structural changes, however, have not been correlated with a possible role. At brain interfaces, astrocytes are capable of acting similar to innate immune cells, with a repertoire of immune processes and factors which are involved in neural tissue protection and maintenance [44,145,146,147,148], together with microglia [149,150]. At least at the BBB, astrocytes dynamically tighten the barrier properties of endothelial cells, and several physiological and pathological conditions may open the barrier [151], including the degeneration of noradrenergic fibers [152].

Astrocytes are equipped with a specialized repertoire of immune receptors which enable them to detect and respond to pathogenic insults and inflammatory stimuli [153,154,155]. Moreover, they produce, secrete, and respond to several cytokines, chemokines, and immune mediators, orchestrating the recruitment and activation of microglia and peripheral innate and adaptive immune cells [156] and have been suggested to play the role of atypical (i.e., inducible) antigen-presenting cells [148].

At rest, astrocytes and microglia cooperate in synaptic plasticity: e.g., astrocytic TGFbeta instructs microglia to prune tagged synapses [150]. Similar to immune cells, however, astrocytes are able to activate upon insult and change their structure and behaviour. Key aspects of astrocyte–microglia crosstalk in neuroinflammation have been comprehensively summarized in recent review articles [104,150,157].

Upon activation, astrocytes become reactive and significantly change their morphology, function, and gene expression. Reactive astrocytes have been classified into two distinct subtypes, neurotoxic (A1), which produce proinflammatory cytokines and ROS, and neuroprotective (A2), which are involved in the clearance of cellular debris, glial scar formation, and BBB repair [158]. However, as is often the case in the immune system, astrocytic reactive response is both heterogeneous and context dependent [159,160,161], being dependent (among other things) on the type of initiating stimuli, CNS locations, or aging. For example, upon systemic stimulation with lipopolysaccharide (LPS) a CXCL10 expressing reactive astrocyte population responsive to interferon (IRRA) appears in regions adjacent to the ventricles, associated with points of entry into the CNS, namely vasculature, ventricles, and the parenchymal surface [162]. This population could play a role in DCN, and it will be interesting to observe the auditory effects of interferon gamma on it, given that a human study has found that this cytokine is decreased in the blood of tinnitus patients [163]. However, it has to be kept in mind that only single-cell transcriptomic experiments would give an idea of the changes due to a given cytokine cocktail, since the population response of astrocytes is very complex and displays several subclasses [159,162]. For example, in physiological conditions, interferon gamma secreted by meningeal NK cells licensed in the gut acts on a subpopulation of astrocytes which secrete the proapoptotic factor TRAIL to block the entry of T cells into the brain parenchyma [164]. This brings up microbiota health as a further modulating factor in the neuroimmune responses and makes it impossible to predict a priori the effect of a particular cytokine on astrocyte functions. Further work is needed to characterize astrocyte subclasses and their interactions in the auditory system. An interesting starting point would be TNF-alpha signaling, since a mouse study showed that this cytokine in the cortex appears causally related to noise-induced tinnitus [29]; although in this study TNF-alpha was found to be produced by microglia, this cytokine can disrupt gap junctions in astrocyte networks [165], thus changing their signaling.

## 7. Conclusions—Astrocytes and Tinnitus

Although no studies have, as yet, investigated the possible roles of astrocytes in tinnitus, the previous sections have shown astrocyte-dependent mechanisms that could likely be important for this disease. These mechanisms fall into three groups: the known roles of astrocytes in tinnitus comorbidities, the known roles of astrocytes in inflammation [166,167], and the emerging role of astrocytes as regulators of neuronal context [133]. As regards the first group, both tinnitus and astrocytes have been associated with disorders of mood [168,169], attention [170], sleep [171,172], and thyroid [173,174], and imaging studies found tinnitus-associated anomalies in auditory, limbic, DMN, and attention circuits plus the cerebellum [3]. As regards the second group, experiments are under way to characterize the inflammatory “signature” of tinnitus and inflammatory responses of the auditory system [28] and, as regards the third group, knowledge gained in other brain regions could be integrated into the current tinnitus models. Understanding the biological correlates of theoretical tinnitus models will yield an important novel target for developing therapeutic strategies for tinnitus.

## Figures and Tables

**Figure 1 brainsci-14-01213-f001:**
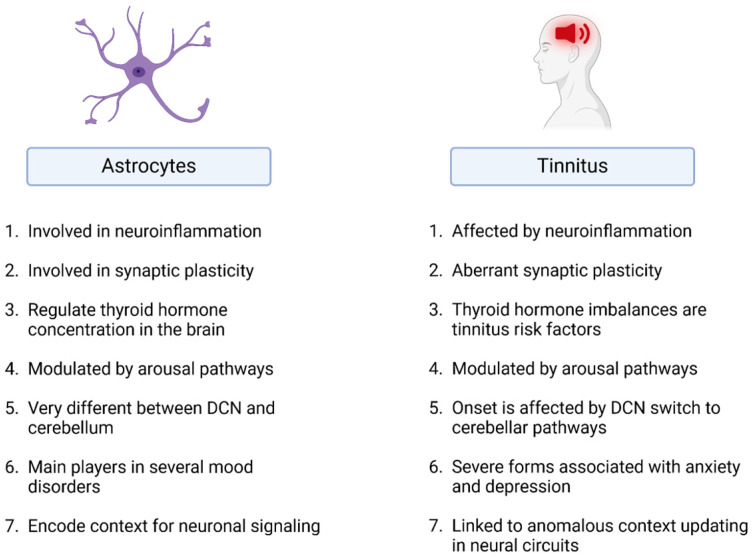
Astrocyte properties known to affect tinnitus. References are given in the main text. Created in Biorender.

**Figure 2 brainsci-14-01213-f002:**
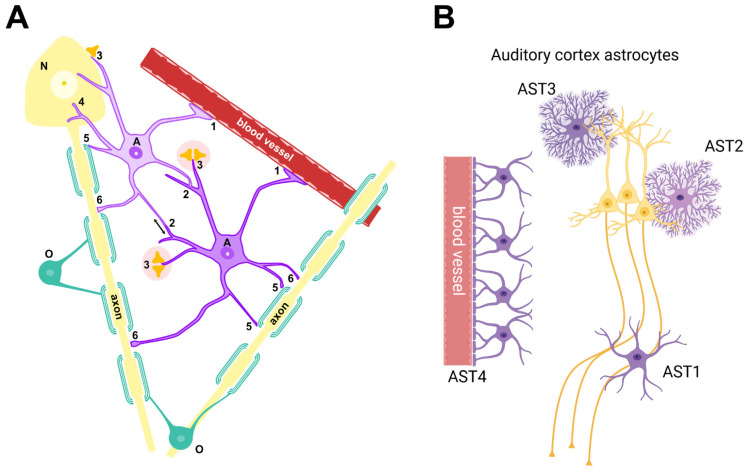
(**A**) Astrocyte connectivity. Astrocytes (A) connect to neurons (N), oligodendrocytes (O), blood vessels, and other astrocytes. Astrocyte endfeet covering blood vessels (1) are involved in maintaining the blood–brain barrier. Astrocytes communicate to each other through an extensive network of gap junctions connecting their terminal branches (2). Very thin peripheral astrocyte processes (PAPs, 3) surround synapses and modulate their functions. Other processes reach neuronal soma (4) and transport nutrients such as glucose or lactate, or neuronal axons at Ranvier’s nodes (6), and help regulate K+. Finally, astrocytes also contact oligodendrocytes (5), therefore affecting myelin homeostasis. A single astrocyte may perform several different functions, but recent research suggests separate subsets of astrocytes specialize for one or few of these. Modified from Biorender. (**B**) Astrocyte subtypes found by single-cell transcriptomic in the rat auditory cortex. The AST1 subtype is associated with myelin and axons, AST2 and AST3 express proteins related to synaptic modulation, and AST4 is involved in neurovascular unit maintenance [49]. Created in Biorender.

**Figure 3 brainsci-14-01213-f003:**
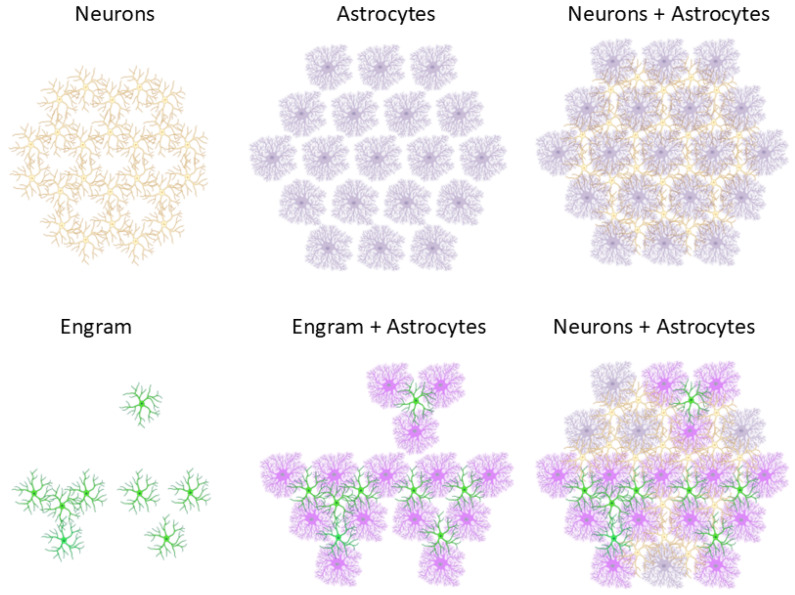
Astrocytes (purple) form a network in parallel to neurons (yellow), whereby a single astrocyte may contact multiple neurons, possibly equalizing plasticity parameters among the neuronal population it contacts. Neurons involved in long-term memory (collectively defined as the ***engram***) change their transcription profiles and can be selectively labelled during memorization (green neurons). Astrocytes surrounding the engram neurons (magenta) also change their transcription profiles and may propagate response changes outside the neuronal engram. Created in Biorender.

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
