# Peer review of "Astrocytes and Tinnitus"

_brainsci, 2024, doi:10.3390/brainsci14121213_

Round 1
Reviewer 1 Report
Comments and Suggestions for Authors
Astrocytes and tinnitus
Paola Perin and Roberto Pizzala
General comments
The review on astrocytes is interesting. I regret, however, the digression at the beginning of the article on the Bayesian theory of tinnitus. I must admit that I disagree with this theory that is so speculative! On my opinion, authors don’t need this hypothesis in this article.
I would focus on how astrocytes can modulate neural activity and what we know about inflammation, in particular after noise trauma or hearing loss.
Recent studies in the auditory system showed a downregulation of KCC2 cotransporters after hearing loss (cochlear nerve section and noise trauma). KCC2 may be controlled by microglia but also astrocytes (Tomoya Kitayama. The Role of Astrocytes in the Modulation ofK+-Cl−-Cotransporter-2 Function. Int J Mol Sci. 2020 Dec 15;21(24):9539. doi: 10.3390/ijms21249539). The authors may want to develop these aspects.
Abstract and main text:
I wont say that the “The main current theory on tinnitus, based on hierarchical predictive coding, suggests that chronic tinnitus derives from changes in brain expectation related to the absence of external sounds.” There are other hypotheses. Let me reverse the question: what makes the author believe it is the main current theory?
What about chronic pain? Is chronic pain also related to this kind of hypothesis? The theory may exist but I don’t think it’s broadly accepted.
“Therefore, if priors define the current spike pattern as “silence”, no sound is perceived.”
Many subjects hear something in deep silence (normal spontaneous activity?) while they don’t hear this activity in normal environment (light low frequency noise). What makes the difference between the two situations in terms of the Bayesian hypothesis?
“fMRI BOLD signal, which reflects neurovascular coupling, may be affected by astrocyte activation even without changes in neuronal activity [Takata et al. 2018], and voxel-based morphometry, which quantifies differences in brain region volumes between pathological or physiological states [Helms 2016], may variably reflect the volume changes of several cell types (including a prominent contribution by astrocytes) in humans [Di Biase et al. 2022] and animal models [Schmidt et al. 210 2021]. “
Very interesting. This point deserves an entire article!
Author Response
-The review on astrocytes is interesting. I regret, however, the digression at the beginning of the article on the Bayesian theory of tinnitus. I must admit that I disagree with this theory that is so speculative! On my opinion, authors don’t need this hypothesis in this article.
We have reduced the emphasis to tinnitus models and removed the digression on bayesian theory.
I would focus on how astrocytes can modulate neural activity and what we know about inflammation, in particular after noise trauma or hearing loss.
Recent studies in the auditory system showed a downregulation of KCC2 cotransporters after hearing loss (cochlear nerve section and noise trauma). KCC2 may be controlled by microglia but also astrocytes (Tomoya Kitayama. The Role of Astrocytes in the Modulation ofK+-Cl−-Cotransporter-2 Function. Int J Mol Sci. 2020 Dec 15;21(24):9539. doi: 10.3390/ijms21249539). The authors may want to develop these aspects.
We have added a discussion of these aspects
I wont say that the “The main current theory on tinnitus, based on hierarchical predictive coding, suggests that chronic tinnitus derives from changes in brain expectation related to the absence of external sounds.” There are other hypotheses. Let me reverse the question: what makes the author believe it is the main current theory?
You are right. We started looking into astrocytes and their possible roles in tinnitus after finding a paper linking them to prior updating; however, you are right that no theory so far is explaining all tinnitus aspects and astrocyte properties would be interesting to investigate no matter what theory is considered, for reasons that we hope to have better explained now
What about chronic pain? Is chronic pain also related to this kind of hypothesis? The theory may exist but I don’t think it’s broadly accepted.
In the discussion of KCC2 we do mention chronic pain as a similar condition
“Therefore, if priors define the current spike pattern as “silence”, no sound is perceived.”
Many subjects hear something in deep silence (normal spontaneous activity?) while they don’t hear this activity in normal environment (light low frequency noise). What makes the difference between the two situations in terms of the Bayesian hypothesis?
We removed the section. However, I think Bayesian model would not be in contrast with this, since it only regards plastic changes. Hearing sounds in deep silence could be explained more simply by reversible gain changes in the auditory system, maybe?
“fMRI BOLD signal, which reflects neurovascular coupling, may be affected by astrocyte activation even without changes in neuronal activity [Takata et al. 2018], and voxel-based morphometry, which quantifies differences in brain region volumes between pathological or physiological states [Helms 2016], may variably reflect the volume changes of several cell types (including a prominent contribution by astrocytes) in humans [Di Biase et al. 2022] and animal models [Schmidt et al. 210 2021]. “
Very interesting. This point deserves an entire article!
Next one :)
Reviewer 2 Report
Comments and Suggestions for Authors
This is an interesting and well-written review of an important topic that is original and thought-provoking. It is concise and keeps to the topic. The authors seem to have an excellent and well-rounded knowledge of recent research into the various types of astrocytes in the brain and their important role in responding to inflammation and other types of neural pathology. Their description of tinnitus mechanisms is rather spartan but this seems appropriate and their description is well balanced. I thought some of their statements about tinnitus were a bit misleading and I have made specific points about minor corrections below. These should be dealt with quite easily and I have no hesitation in recommending this article for publication. It gives a detailed and timely review of the roles of astrocytes that may be of relevance to tinnitus and makes a compelling argument that their role should be considered in any comprehensive understanding of the mechanisms that cause tinnitus.
Minor comments:
On line 26 the authors state that, “no condition has been found to always consistently induce tinnitus”. I think this is a bit misleading as high doses of salicylate are thought to reliably generate tinnitus in both animal models and humans (Stolzberg, Salvi and Allman: 2012). Salicylate isn’t a perfect model because of the high doses involved producing a variety of types of ototoxic damage and directly entering the brain, but it is reliable. Please be more precise.
Related to this confusion is the statement on line 28, “among individuals exposed to the same stimuli, some develop tinnitus and some don’t”. This is primarily true of humans exposed to occupational noise and animals that have been acoustically over-exposed. I think the term stimuli is rather vague and that the authors should specify what type of stimuli have been implicated in producing tinnitus in a subgroup of animals or humans.
There are two broad groups of models that seek to explain tinnitus – a bottom-up and a top-down group and I think the authors should make it clearer that the hierarchical predictive coding (discussed in lines 33 - 36) primarily applies to the top-down models. If you record from the inner hair cells or auditory nerve there is a very faithful coding of the sounds that have been transduced. There is very little involvement of predictive coding in the more peripheral parts of the auditory system as this is much more relevant at thalamocortical levels.
In line 36 they state, “only unexpected signals are carried through” but I think this only applies to background noise which is not being attended to. If you are attending to speech or music then the expected sounds are allowed through. Please clarify.
In Figure 2, I can’t see any labelling of parts “A” and “B”.
In line 153 the authors state that the DCN is, “the first structure changing its activity in the presence of tinnitus”. However, I don’t think it is clear what is meant by this and they would be better to omit or at least modify this statement. Do they mean this in terms of a temporal sequence where noise exposure affects neural structures? If so I would argue that the cochlea is the first structure to show changes. Do they mean the first central structure that is affected? If so I would argue that the cochlear root nucleus is the first to be affected and that the ventral cochlear nucleus is also thought to be involved in tinnitus development and is at the same hierarchical level as the DCN.
Line 298 there are two “ls” in Schilling et al., 2023
Author Response
On line 26 the authors state that, “no condition has been found to always consistently induce tinnitus”. I think this is a bit misleading as high doses of salicylate are thought to reliably generate tinnitus in both animal models and humans (Stolzberg, Salvi and Allman: 2012). Salicylate isn’t a perfect model because of the high doses involved producing a variety of types of ototoxic damage and directly entering the brain, but it is reliable. Please be more precise.
Thanks for the comment. We have corrected the paragraph, which now reads:
In animal models, several standard procedures are used to induce tinnitus, such as noise trauma and high-dose salicylate [Galazyuk and Brozoski 2020] but concurring hearing disturbances, such as hyperacusis and hearing loss, may act as confounding factors regarding what is being measured [Eggermont 2013, Galazyuk and Brozoski 2020, Wake et al. 2024]; moreover, the measured functional brain changes correlated to tinnitus vary depending on the inducing procedure [Noreña et al. 2010, Salvi et al. 2021]. This suggests the presence of a major gap in our understanding of tinnitus.
Related to this confusion is the statement on line 28, “among individuals exposed to the same stimuli, some develop tinnitus and some don’t”. This is primarily true of humans exposed to occupational noise and animals that have been acoustically over-exposed. I think the term stimuli is rather vague and that the authors should specify what type of stimuli have been implicated in producing tinnitus in a subgroup of animals or humans.
We have removed the statement and rewritten the paragraph as shown in the point above.
There are two broad groups of models that seek to explain tinnitus – a bottom-up and a top-down group and I think the authors should make it clearer that the hierarchical predictive coding (discussed in lines 33 - 36) primarily applies to the top-down models. If you record from the inner hair cells or auditory nerve there is a very faithful coding of the sounds that have been transduced. There is very little involvement of predictive coding in the more peripheral parts of the auditory system as this is much more relevant at thalamocortical levels.
We have changed the chapter to include different tinnitus models
In line 36 they state, “only unexpected signals are carried through” but I think this only applies to background noise which is not being attended to. If you are attending to speech or music then the expected sounds are allowed through. Please clarify.
We have removed most of the Bayesian model discussion, as per Reviewer 1 request.
In Figure 2, I can’t see any labelling of parts “A” and “B”.
We have added labels.
In line 153 the authors state that the DCN is, “the first structure changing its activity in the presence of tinnitus”. However, I don’t think it is clear what is meant by this and they would be better to omit or at least modify this statement. Do they mean this in terms of a temporal sequence where noise exposure affects neural structures? If so I would argue that the cochlea is the first structure to show changes. Do they mean the first central structure that is affected? If so I would argue that the cochlear root nucleus is the first to be affected and that the ventral cochlear nucleus is also thought to be involved in tinnitus development and is at the same hierarchical level as the DCN.
We have corrected the statement to "The DCN has been found in animal studies to be central for tinnitus onset"
Line 298 there are two “ls” in Schilling et al., 2023
Corrected, thanks.